# Debiasing Vision-Language Models via Biased Prompts

### Abstract

Machine learning models have been shown to inherit biases from their training datasets. This can be particularly problematic for vision-language foundation models trained on uncurated datasets scraped from the internet. The biases can be amplified and propagated to downstream applications like zero-shot classifiers and text-to-image generative models. In this study, we propose a general approach for debiasing vision-language foundation models by projecting out biased directions in the text embedding. In particular, we show that debiasing only the text embedding with a calibrated projection matrix suffices to yield robust classifiers and fair generative models. The proposed closed-form solution enables easy integration into large-scale pipelines, and empirical results demonstrate that our approach effectively reduces social bias and spurious correlation in both discriminative and generative vision-language models without the need for additional data or training.

## 1 Introduction

Foundation vision-language models, such as CLIP (Radford et al., 2021), DALLE-2 (Ramesh et al., 2022), Imagen (Saharia et al., 2022), and Stable Diffusion (Rombach et al., 2022), which are trained on extensive multimodal data at a massive scale, have led to a significant shift in the landscape of machine learning systems. Specifically, contrastive vision-language encoders like CLIP have the ability to perform zero-shot inferences without fine-tuning, and language embeddings can be used to train high-quality text-to-image models (Rombach et al., 2022).

While vision-language models demonstrate impressive capabilities, it is important to recognize that they may also exacerbate biases (Mehrabi et al., 2021; Agarwal et al., 2021; Wang et al., 2021; Cho et al., 2022). Recent studies (Birhane et al., 2021) have shown that the datasets these models are trained on can contain inappropriate image-text pairs with stereotypes, racist content, and ethnic slurs. The biases are then propagated to downstream applications (Agarwal et al., 2021; Wang et al., 2021), resulting in biased predictions. In addition to social biases, zero-shot models derived from vision-language models can also suffer from more general forms of spurious correlations such as image background, leading to poor group robustness (Zhang & Ré, 2022). Biases also exist in generative models, where generated images may exhibit bias towards certain genders and races (Cho et al., 2022; Mishkin et al., 2022). Substantial progress has been made recently toward mitigating biases in vision-language models (Parraga et al., 2022; Berg et al., 2022; Zhang & Ré, 2022). However, many current approaches for addressing bias in models require training or fine-tuning the models using resampled datasets or modified objectives, which can be computationally intensive for foundation models.

In this work, we propose a general approach for self-debiasing foundation vision-language models by projecting out biased directions in the text embedding. Given a vision-language encoder such as CLIP, we define a set of biased directions in the embedding using prompts that describe the biases. For instance, prompts like "a photo of a male/female" define a biased subspace in the latent space. One approach to mitigating these biases is to construct a projection matrix, a linear transformation of the text embedding that projects out the biased directions (Bolukbasi et al., 2016). However, solely relying on prompts to define biased directions may be unstable and noisy (Gonen & Goldberg, 2019). To address this issue, we propose a calibration loss that minimizes the discrepancy of a pair of prompt embeddings. For example, given a projection matrix that removes gender information, the projected vectors of prompts "a photo of a male doctor" and "a photo of a

female doctor" should be similar. Based on this principle, we design an objective to calibrate the projection matrix, which has an easily solvable closed-form solution. This allows for the construction of the projection matrix to be *training-free and requires no downstream dataset or labels*, making it suitable for large-scale models. Empirically, we find that debiasing only the text embedding with a calibrated projection matrix suffices to improve the group robustness of zero-shot models on well-established benchmarks.

We then extend our approach to generative models such as Stable Diffusion (Rombach et al., 2022), a widely adopted text-to-image model conditioned on text embeddings from CLIP (Radford et al., 2021). The inherent challenge lies in the fact that generative models are distinctly dissimilar from zero-shot classification, where the target classes are explicitly defined. With generative models, our objective is to develop a debiasing matrix that is universally applicable to every prompt. This matrix can then be employed as a standard preprocessing stage prior to feeding the text embedding into the generative model. To accomplish this, we solve the calibration matrix with a set of positive pairs which comprise various prompts from the training dataset, and debias the unseen prompts with the obtained matrix. Similar to debiasing zero-shot models, the projection matrix improves the diversity of generated images from text-to-image models without altering the model parameters.

In short, this work makes the following contributions:

- We present a simple and general approach for debiasing vision-language models;
- The proposed approach does not require training, data, or labels, making it computationally efficient for use with foundation models;
- We evaluate our approach through experiments on both discriminative (zero-shot, text-image retrieval) and generative (diffusion) vision-language models.

## 2 Related Works

Vision-Language models (Radford et al., 2021; Ramesh et al., 2022; Saharia et al., 2022; Rombach et al., 2022) have become increasingly widespread in recent years. However, these models are known to suffer from spurious correlations and can be biased towards certain races and gender. Birhane et al. (2021) study the datasets these models are trained on and show that the data biases can be inherited by the models. Various methods have been proposed to address biases, but many of them only address single-modality models.

**Biases in Language Models** Large-scale language models have been shown to contain harmful or misrepresentative biases (Blodgett et al., 2020; Nadeem et al., 2020; Weidinger et al., 2021). Previous research has demonstrated the presence of gender bias in natural language processing systems (Bolukbasi et al., 2016; Zhao et al., 2019) as well as racial bias (Manzini et al., 2019; Garg et al., 2018). Bolukbasi et al. (2016) first proposed the use of orthogonal projections to remove gender biases in word embeddings. This approach was later extended to debiasing sentence embeddings (Liang et al., 2020). Alternative methods include regularizing the models with constraints on training data (Zhao et al., 2017; Huang et al., 2019) or directly modifying the dataset (Sun et al., 2019; Zhao et al., 2019). However, scaling these approaches to large foundation models can be challenging as they often require retraining the backbone encoders.

**Biases in Vision Models** Gender and racial biases have also been widely explored in computer vision (Alvi et al., 2018; Wang & Deng, 2020), for discriminative models (Wang et al., 2019) and generative models (Xu et al., 2018; Grover et al., 2019; Cho et al., 2022). Many debiasing approaches aim to learn good representations via adversarial training (Madras et al., 2018; Wang et al., 2020), or augmenting the biased dataset (Ramaswamy et al., 2021; Chuang & Mroueh, 2021). Beyond social bias, many works study spurious correlations, a more general form of bias that can include features such as image background or other non-target attributes that are correlated with labels. This problem of spurious correlations is often studied and tackled as a group robustness problem (Sagawa et al., 2019; Izmailov et al., 2022). SJ: there s probably more work on spurious correlation, look e.g. at Josh's shortcuts paper Kirichenko et al. (2022) show that last layer re-training is sufficient for robustness to spurious correlations, which aligns with our finding that debiasing the zero-shot weights suffices to yield robust classifiers.

**Biases in Vision-Language Models** Recently, biases in multimodal settings have gained significant attention (Agarwal et al., 2021; Hall et al., 2023). Wang et al. (2021) propose to remove dimensions in the CLIP embedding that are highly correlated with gender attributes. Berg et al. (2022) debias the CLIP models with prompt learning via an adversarial approach. Seth et al. (2023) learn additive residual image representations to offset the biased representations. Recently, Zhang & Ré (2022) address the group robustness of vision-language models with contrastive learning. These previous works are data-oriented, where models are trained or finetuned on labeled datasets. In contrast, our approach is fully zero-shot, which does not require any downstream dataset and model training. To debias generative models, a recent work (Friedrich et al., 2023) pre-defines a look-up table to provide fair guidance for text-to-image diffusion models. Nevertheless, this method encounters limitations when faced with previously unseen classes that are absent from the look-up table and cannot be applied to discriminative models, while our approach generalizes well to new concepts. There is another line of work focusing on removing malignant concepts in generative models. Schramowski et al. (2023) advocate for steering the diffusion process away from inappropriate content by subtracting the text embedding with unsafe guidance. Subsequent studies (Gandikota et al., 2023; Kumari et al., 2023) have built upon this idea by selectively fine-tuning parameters within these generative models. In contrast to these efforts, our methodology aims to balance group proportions within the output distribution. Notably, our technique is versatile enough for implementation in both discriminative and generative models and is characterized by its computational efficiency.

## 3 Biases and Spurious Correlations

We consider a dataset where each input $x \in \mathcal{X}$ is associated with multiple attributes, including the target class $y \in \mathcal{Y}$ and a spurious attribute $a \in \mathcal{A}$. We focus on the case where biases are present and the attribute $a$ is spuriously correlated with the label $y$. For instance, the class "doctor" could be correlated with the spurious attribute "gender" in the datasets foundation models are trained on (Birhane et al., 2021). Importantly, these biases can be transferred to downstream tasks, both discriminative and generative.

**Discriminative Models** First, we examine the biases present in zero-shot classifiers obtained via a vision-language encoder such as CLIP. These classifiers are built by assigning each row of the linear classifier weight $\beta \in \mathbb{R}^{K \times d}$ to be the embedding of a "class prompt", for example, "a photo of a [class name]" (Radford et al., 2021). Importantly, it does not require any data or training to construct these zero-shot classifiers. However, it is possible for these zero-shot classifiers to inherit biases from the dataset used to train the vision-language models. To study these biases, we utilize the group robustness framework proposed by Sagawa et al. (2019). In this setting, groups are defined by a combination of the labels and spurious attributes: $\mathcal{G} \in \mathcal{Y} \times \mathcal{A}$. Given a distribution $P_g$ conditioned on $g \in \mathcal{G}$ and a loss function $\ell : \mathcal{Y} \times \mathcal{Y} \to \mathbb{R}$, group robustness requires that the classifier $f : \mathcal{X} \to \mathcal{Y}$ achieves a small gap between its worst-group error and average error:

$$\max_{g \in \mathcal{G}} \mathbb{E}_{x,y \sim P_g} \left[ \ell(f(x), y) \right] - \mathbb{E}_{x,y \sim P} \left[ \ell(f(x), y) \right]. \tag{1}$$

Our goal is to attain superior group robustness, namely, minimizing the discrepancy in error rates. The definition of metrics for text-image retrievals, such as maximal skewness (Geyik et al., 2019), will be deferred to the experiment section.

**Generative Models** A text-to-image model learns a conditional distribution $\hat{P}(X|Z = z)$, where $z$ is the embedding of the prompt. However, the biased nature of the dataset used to train the generative model can affect the distribution $\hat{P}$. To measure the bias present in generative models, recent works (Choi et al., 2020; Teo & Cheung, 2021) propose using statistical parity. Specifically, given a classifier $h : \mathcal{X} \to \mathcal{A}$ for the spurious attribute, the discrepancy of the generative distribution $\hat{P}$ is defined as the L2 norm between empirical and uniform distributions (Choi et al., 2020):

$$\sqrt{\sum_{a \in \mathcal{A}} \left( \mathbb{E}_{x \sim \hat{P}} \left[ \mathbb{1}_{h(x)=a} \right] - 1/|\mathcal{A}| \right)^2}. \tag{2}$$

In practice, the expectation is estimated empirically with samples. A fair generative model minimizes the discrepancy by ensuring that each attribute $a \in \mathcal{A}$ has an equal probability of occurring in the data.

|  | CelebA | | Waterbird | |
| --- | --- | --- | --- | --- |
|  | Male | Female | Land | Water |
| dark hair / landbird | 0.83 | 0.78 | 0.75 | 0.66 |
| blond hair / waterbird | 0.77 | 0.85 | 0.65 | 0.70 |

**Table 1: Cosine similarity between classifier weights and spurious directions:** in both datasets, the classifier weights are biased toward certain spurious attributes. The labels and spurious attributes are binary variables in both datasets.

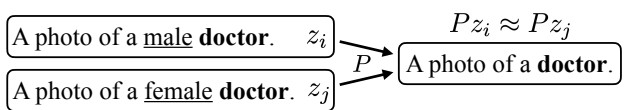

**Figure 1: Calibration with Positive Pairs.** Upon projecting out irrelevant features (such as gender), the embeddings of group prompts ($z_i$ and $z_j$) should exhibit similarity and contain only information pertaining to the target class (e.g. doctor).

## 4 Debiasing Discriminative Models

It is essential for a robust classifier to evade dependence on irrelevant features present in images. This necessitates the classifier to be invariant to image backgrounds and insensitive to attributes such as race or gender. Prior research has employed datasets with target labels and spurious attributes to quantify and eliminate biases (Sagawa et al., 2019; Zhang & Ré, 2022). However, this approach is not feasible in a zero-shot setting, where data and training are prohibitive.

### 4.1 Measuring Biases with Prompts

In contrast to previous approaches, our proposed method for measuring biases utilizes prompts, drawing inspiration from studies on debiasing word embeddings (Bolukbasi et al., 2016). The use of vision-language contrastive training allows for the description of irrelevant features through natural language. As such, embeddings of prompts such as "a photo of a [irrelevant attribute]" can capture these spurious features in the visual embedding. Consequently, the bias of a classifier can be quantified by computing the cosine similarity between its weights and the corresponding spurious feature. Table 1 illustrates the cosine similarity between the embeddings of prompts that describe the target classes and irrelevant attributes, using two popular group robustness benchmarks: Waterbird (Sagawa et al., 2019) and CelebA (Liu et al., 2015). The details of datasets and the specific prompts can be found in section 5 and appendix C.1. The results demonstrate that the classifier weights are inclined towards certain irrelevant attributes (gender or image background), implying that the classifiers could use these spurious directions to make predictions.

### 4.2 Debiasing via Orthogonal Projection

As the zero-shot weights can also be viewed as natural language embeddings, a straightforward approach is to follow the debiasing pipeline employed in word and sentence embeddings (Bolukbasi et al., 2016; Liang et al., 2020). In particular, to make the classifier invariant to irrelevant features, we align the classifier weights with the orthogonal complement of the embeddings associated with spurious prompts. Let $A \in \mathbb{R}^{d \times m}$ be a matrix whose columns are the embeddings of spurious prompts. The orthogonal projection matrix is then:

$$P_0 = I - A(A^T A)^{-1} A^T.$$

We can use the projection matrix to eliminate spurious directions in a text embedding $z$ as $P_0 z$.

### 4.3 Calibrating the Projection Matrix

It is essential to acknowledge that the estimation of the irrelevant feature directions may introduce an approximation error in the projection matrix (Gonen & Goldberg, 2019). Additionally, in certain scenarios, it may be challenging to thoroughly describe the irrelevant attribute using a limited number of prompts, resulting in increased uncertainty in the projection matrix estimation. This issue is also evident in our empirical results (Table 2 and 4), where the use of orthogonal projection fails to enhance performance.

To improve the estimation of the projection matrix, we propose to leverage *positive pairs* of prompts that are expected to have the same semantic meaning after projection. In particular, the embedding of prompts such

as "a photo of a [class name] with [spurious attribute]" should only contain information about "[class name]" after projecting out the spurious information, as Figure 1 illustrates. Motivated by this intuition, we propose to regularize the difference between the projected embeddings using a set of positive pairs $\mathcal{S}$:

$$\min_P \ \|P - P_0\|^2 + \frac{\lambda}{|\mathcal{S}|} \sum_{(i,j) \in \mathcal{S}} \|Pz_i - Pz_j\|^2, \tag{3}$$

where $(z_i, z_j)$ is the embedding of pair $(i, j)$ in $\mathcal{S}$ and $(i, j)$ are prompts that describe the same class but different spurious attributes. The loss encourages the linear projection $P$ to be invariant to the difference between $(i, j)$, i.e., the spurious attributes. The optimization problem has a convenient closed-form solution, as demonstrated in Lemma 4.1.

**Lemma 4.1.** *The minimizer of the calibration loss is*

$$P^* = P_0 \Big( I + \frac{\lambda}{|\mathcal{S}|} \sum_{(i,j) \in \mathcal{S}} (z_i - z_j)(z_i - z_j)^T \Big)^{-1}.$$

We can obtain an interpretation of the minimizer by relating it to singular value decomposition (SVD). Let $Z_{\text{diff}} \in \mathbb{R}^{d \times |S|}$, where the columns of $Z_{\text{diff}}$ enumerate the pairwise difference $z^i - z^j$ for all $(i, j) \in S$. The matrix $Z_{\text{diff}}$ defines a subspace that represents the variation in the embedding when the irrelevant feature is changed. Using $Z_{\text{diff}}$, the minimizer can be written as $P^* = P_0(I + \lambda' Z_{\text{diff}} Z_{\text{diff}}^T)^{-1}$, where we define $\lambda' = \lambda/|\mathcal{S}|$ to simplify the notation. Assume that the SVD of $Z_{\text{diff}}$ is $U \Sigma V^T$. Then we have $Z_{\text{diff}} Z_{\text{diff}}^T = U \Sigma^2 U^T$. The optimal solution $P^*$ can then be rewritten as

$$P^* = P_0 (U(I + \lambda' \Sigma^2) U^T)^{-1} = P_0 \underbrace{U (I + \lambda' \Sigma^2)^{-1} U^T}_{\text{Calibration Matrix}}.$$

We can see that $U(I + \lambda' \Sigma^2)^{-1} U^T$ acts as a calibration term. Before multiplying the text embedding with the projection matrix $P_0$, variation due to the change of the spurious feature, namely, the eigenvectors with large squared singular value in $Z_{\text{diff}}$ (spurious direction) will be down-weighted due to the inverse $(I + \lambda' \Sigma^2)^{-1}$. Therefore, varying the spurious attributes should result in similar embeddings after multiplying the calibration matrix.

### 4.4 Relation to an Equalization Loss

Finally, we provide an equivalent form of the calibrated projection. Ideally, we want each row of the classifier weight $\beta \in \mathbb{R}^{K \times d}$ to have similar cosine similarity to pairs of embeddings in $\mathcal{S}$. For instance, the embedding of "a photo of a doctor" should be equally similar to "a photo of a male doctor" and "a photo of a female doctor". In this section, we will show that the optimum of the calibration loss does satisfy this criterion.

We consider the following objective for obtaining a debiased text embedding $z \in \mathbb{R}^d$ of a prompt given its initialization $z_0 \in \mathbb{R}^d$ from the text encoder:

$$\min_z \ \|z - z_0\|^2 + \frac{\lambda}{|\mathcal{S}|} \sum_{(i,j) \in \mathcal{S}} (z^T z_i - z^T z_j)^2. \tag{4}$$

The loss encourages the embedding $z$ to have similar cosine similarity to embeddings in positive pairs while maintaining proximity to the initialization $z_0$. Objective (4) has the same optimal solution as the calibration loss (3).

**Lemma 4.2.** *The minimizer of objective (4) reads*

$$z^* = \underbrace{\Big( I + \frac{\lambda}{|\mathcal{S}|} \sum_{(i,j) \in \mathcal{S}} (z_i - z_j)(z_i - z_j)^T \Big)^{-1}}_{\text{Calibration Matrix}} z_0$$

*In particular, we have $P_0 z^* = P^* z_0$ where $P^*$ is the minimizer of the calibration loss (3).*

| Backbone | CLIP ResNet-50 | | | | | | CLIP ViT-L/14 | | | | | |
|---|---|---|---|---|---|---|---|---|---|---|---|---|
| Dataset | **Waterbird** | | | **CelebA** | | | **Waterbird** | | | **CelebA** | | |
| | WG | Avg | Gap | WG | Avg | Gap | WG | Avg | Gap | WG | Avg | Gap |
| *methods using data and labels* | | | | | | | | | | | | |
| ERM Linear | 7.9 | 93.5 | 85.6 | 11.9 | 94.7 | 82.8 | 65.9 | 97.6 | 31.7 | 28.3 | 94.7 | 66.4 |
| ERM Adapter | 60.8 | 96.0 | 35.2 | 36.1 | 94.2 | 58.1 | 78.4 | 97.8 | 19.4 | 36.7 | 94.2 | 57.5 |
| WiSE-FT | 49.8 | 91.0 | 41.2 | 85.6 | 88.6 | 3.0 | 65.9 | 97.6 | 31.7 | 80.0 | 87.4 | 7.4 |
| DFR (Sub) | 63.9 | 91.8 | 27.9 | 76.9 | 92.5 | 15.6 | 51.9 | 95.7 | 43.8 | 76.3 | 92.1 | 15.8 |
| DFR (Up) | 51.3 | 92.4 | 41.1 | 89.6 | 91.8 | 2.2 | 65.9 | 96.1 | 30.2 | 83.7 | 91.2 | 7.5 |
| CA | 83.7 | 89.4 | 5.7 | 90.0 | 90.7 | 0.7 | 86.9 | 96.2 | 9.3 | 84.6 | 90.4 | 5.8 |
| *methods without data and labels* | | | | | | | | | | | | |
| Zero-shot | 39.6 | 77.3 | 37.7 | 75.9 | 82.3 | 6.4 | 45.3 | 84.4 | 39.1 | 72.8 | 87.6 | 14.9 |
| Orth-Proj (Ours) | 48.1 | 83.6 | 35.4 | 61.4 | 86.4 | 25.0 | 61.4 | 86.4 | 25.0 | 71.1 | 87.0 | 15.9 |
| Orth-Cali (Ours) | **74.0** | 78.7 | **4.7** | **82.2** | 84.4 | **2.2** | **68.8** | 84.5 | **15.7** | **76.1** | 86.2 | **10.1** |

**Table 2: Group Robustness of Vision-Language Models.** For each backbone, the first blocks contain methods that require data and labels, while the second blocks contain zero-shot methods. The numbers for the first block are adopted from Zhang & Ré (2022). The proposed calibration loss achieves comparable or even smaller gaps between average and worst group accuracy without the need for any data or labels.

Lemma 4.2 shows that the optimal solution of (4) is equivalent to multiplying the original embedding $z$ with the calibration matrix defined before. Applying the projection $P_0$ to $z^*$ leads to the same weight as in Lemma 4.1. The equalization objective has a similar motivation as the equalization step proposed by Bolukbasi et al. (2016) in their work on removing gender bias from word embeddings. Given a set of word embeddings that has the same semantic meaning except for gender, their approach centers these embeddings by simply setting them to the average embedding of the set. After centering, any word in the dictionary will be equidistant to all words in the set. However, their approach fails to extend to debiasing zero-shot classifiers as we are primarily concerned with the embedding of $z$ instead of $z_i$ and $z_j$. Our approach differs in that we modify the embedding of the target prompt $z$, rather than the embedding of positive pairs, making it applicable for debiasing general vision-language models.

## 5 Experiments: Discriminative Models

### 5.1 Group Robustness against Spurious Correlations

By following the setting of Zhang & Ré (2022), we evaluate our approach on two popular benchmarks for evaluating spurious correlations, Waterbird (Sagawa et al., 2019) and CelebA (Liu et al., 2015). On Waterbird, a water/land background is a confounding factor for the waterbirds/landbirds class, while on CelebA the binary gender is the spurious feature for blond/dark hair. As such, both datasets contain four groups defined by the labels and the spurious attributes.

We evaluate our approach against several baselines, including zero-shot classification (Radford et al., 2021), empirical risk minimization (ERM) with linear probing (Kumar et al., 2022), and ERM with non-linear adapter (Gao et al., 2021). Additionally, we also consider three recent methods designed to improve the group robustness of vision-language foundation classifiers:

- Weight Space Ensembling (WiSE-FT) (Wortsman et al., 2022), which trains a linear classifier first using ERM and then combines the classifier outputs with the initial zero-shot predictions;
- Deep Feature Reweighting (DFR) (Kirichenko et al., 2022), which trains a linear probe on embeddings obtained from a pre-trained model using group-balanced data. Following Zhang & Ré (2022), the group labels are replaced with zero-shot predictions;
- Contrastive Adapter (CA) (Zhang & Ré, 2022), which trains adapters using contrastive learning to bring embeddings in the same class closer.

| | Waterbird | | | CelebA | | |
|---|---|---|---|---|---|---|
| | WG | Avg | Gap | WG | Avg | Gap |
| Proj only | 48.1 | 83.6 | 35.4 | 61.4 | 86.4 | 25.0 |
| Cali only | 55.6 | 81.6 | 26.0 | 81.6 | 84.7 | 3.1 |
| Proj + Cali | **74.0** | 78.7 | **4.7** | **82.2** | 84.4 | **2.2** |

Table 4: Dissecting Orthogonal Projections. We evaluate variants of orthogonal projection with ResNet-50 backbone.

| | CLIP ViT-B/32 | | | CLIP ViT-L/14 | | |
|---|---|---|---|---|---|---|
| | Gen | Race | Age | Gen | Race | Age |
| Zero-shot | .206 | .743 | .797 | .206 | .768 | .703 |
| Orth-Proj | .146 | .755 | **.635** | .349 | .605 | .706 |
| Orth-Cali | **.102** | .638 | .641 | **.200** | .461 | **.662** |

Table 5: Measuring biases on FairFace. We MaxSkew@1000 (the smaller the better) on FairFace validation set.

It is important to note that **all of the baselines** except the zero-shot classifier **require at least training data and class labels**, while our debiasing approach does not require access to any input data, labels, or group labels, which follows the principles of zero-shot learning.

We evaluate the performance of our proposed approach using two CLIP backbones: ResNet-50 (He et al., 2016) and ViT-L/14 (Dosovitskiy et al., 2020). The results are presented in Table 2. The results indicate that a simple application of the orthogonal projection (Orth-Proj) by itself only yields limited improvement of the worst group accuracy, whereas the calibration loss (Orth-Cali) significantly improves robustness across datasets and base models. The proposed Orth-Cali method achieves comparable or even smaller gaps between average (Avg) and worst group (WG) accuracy compared to the state-of-the-art contrastive adapter (Zhang & Ré, 2022), without the need for any data or labels. Note that the baselines generally achieve better average accuracy as they require fine-tuning on the target datasets.

| | Waterbird | | | CelebA | | |
|---|---|---|---|---|---|---|
| | WG | Avg | Gap | WG | Avg | Gap |
| $\lambda = 200$ | 71.8 | 80.8 | 9.0 | 80.7 | 83.9 | 3.2 |
| $\lambda = 400$ | 72.9 | 79.5 | 6.6 | 81.6 | 84.2 | 2.6 |
| $\lambda = 600$ | 73.5 | 79.2 | 5.7 | 81.9 | 84.3 | 2.4 |
| $\lambda = 1000$ | 74.0 | 78.7 | 4.7 | 82.2 | 84.4 | 2.2 |

Table 3: Sensitivity to $\lambda$. We vary the weighting parameter $\lambda$ and evaluate group robustness with ResNet-50 backbone.

Empirically, we found that gradually increasing the parameter $\lambda$ improves the worst group accuracy and leads to a stable solution as shown in Table 3. Therefore, for all the experiments on discriminative models, we set $\lambda$ to 1000 by default. To investigate the importance of orthogonal projection and calibration, we present an ablation study in Table 4. The results indicate that the calibration loss alone ($P_0 = I$) performs well on the CelebA dataset, as the spurious feature (gender) is relatively easy to describe with prompts. However, performance drops on the Waterbird dataset without a good initialization from the orthogonal projection. More ablation studies can also be found in Appendix D, where we demonstrate the importance of class names in positive pairs.

## 5.2 Debiased Information Retrieval

Fairness in text-image retrieval has gained increasing attention in recent years. Building on the work of Berg et al. (2022), we propose to utilize the MaxSkew metric, introduced by Geyik et al. (2019), to evaluate the level of fairness in the retrieval results. Specifically, we conduct our analysis on the FairFace dataset (Kärkkäinen & Joo, 2019), which is specifically designed to address issues of fairness in facial recognition systems. Given a ranked list of images in response to a text query, let $r_{a,k}$ be the ratio of the top k images that are labeled with attribute $a$. Then MaxSkew@k is defined as $\max_{a \in \mathcal{A}} \log \frac{r_{a,k}}{1/|\mathcal{A}|}$. It quantifies the maximal discrepancy between the ratio of top k images labeled with a specific sensitive attribute, denoted as $r_{a,k}$, and the uniform weight $1/|\mathcal{A}|$, where $\mathcal{A}$ represents the set of sensitive attributes. The MaxSkew metric provides a useful measure of fairness in text-image retrieval systems, by assessing the degree to which the retrieval results are evenly distributed across sensitive attributes. A small MaxSkew value indicates that the distribution of retrieved images across different sensitive attributes is close to being uniform.

To measure the bias, we query the validation set of FairFace based on 10 prompts that are uncorrelated with facial expressions or sensitive attributes, e.g., "a photo of a [concept] person", where the [concept] is a

neutral concept such as evil or smart. The detailed prompts are described in Appendix C. We measure the MaxSkew based on three labeled attributes of FairFace: gender, race, and age. Table 5 shows the average MaxSkew@1000 over concepts, demonstrating that our approach significantly reduces the MaxSkew across different attributes and backbones.

# 6 Debiasing Generative Models

We now explore the possibility of extending the methodology developed for discriminative models to generative models. Our primary focus is on addressing social group biases, specifically gender and race discrepancy, as measured by metric (1). In particular, our running example and main experiment is to query the generative model using profession-related prompts, specifically "a photo of a [profession]". Empirically, the generated images were found to exhibit a strong bias towards certain gender and race, and we attempt to improve the diversity of generated images with the proposed equalization loss in this section. We also demonstrate that our approach can address spurious correlations beyond social biases.

**A Single Matrix for Comprehensive Debiasing**  Unlike the well-defined targets prevalent in zero-shot classification, the nature of generative models requires a more universal solution. Specifically, we seek to derive a debiasing matrix capable of accommodating any prompt. This matrix could subsequently be treated as a standardized preprocessing step, applied prior to the introduction of the embedding into the generator.

To achieve this, we optimize the equalization loss with positive pairs consisting of an enumeration of "a photo of a [attribute] [profession]" where the [attribute] is a member of the set of gender or races and the [profession] is a job title sampled from a training set. For instance, to mitigate gender bias, we adopt $\mathcal{S} = \{(\text{"a photo of a male doctor"}, \text{"a photo of a female doctor"}), \cdots, (\text{"a photo of a male engineer"}, \text{"a photo of a female engineer"})\}$. By solving the calibration matrix with professions in the training set, we expect the obtained matrix can also mitigate the biases in unseen professions. Note that we optimize the equalization loss (4) without applying the initial orthogonal projection matrix $P_0$. This is because our goal is to balance rather than completely eliminate biased information in the generated images.

# 7 Experiments: Generative Models

To evaluate the effectiveness of our approach in the context of generative models, we conducted experiments using the Stable Diffusion (SD) v2.1 framework (Rombach et al., 2022). We construct a list of professions that consists of 100 job titles with GPT-4 (OpenAI, 2023) and randomly separate them into 80 training and 20 testing professions. The complete list can be found in appendix C. In alignment with the framework proposed by Kärkkäinen & Joo (2019), we consider the gender attributes of male and female, and racial attributes of White, Asian, Black, Indian, and Latino.

It is essential to recognize gender and race are complex social constructs that cannot be simply reduced to binary or discrete categories. The choice of using binary gender and discrete race attributes in our work was primarily based on the existing literature and benchmark datasets that have commonly adopted this setting for evaluation purposes (Barocas et al., 2019).

Evaluating generative models can be challenging without the use of human labels. Inspired by Cho et al. (2022), we used sensitive attribute classifiers to predict the sensitive attributes of the generated images. The discrepancy, as defined in equation 2, was then calculated. In particular, we generate 100 images for each train / test profession for evaluation, resulting in 10000 images for each model. We then leverage the CLIP classifier to predict the sensitive attributes to calculate the discrepancy. An alternative to CLIP is the FairFace classifier (Kärkkäinen & Joo, 2019); however, we found that the domain shift between the FairFace dataset and the generated images significantly impairs its performance. The debiased and biased models share the same random seed for fair comparison. We set $\lambda = 500$ for all the experiments in this section. Note that it only takes a few seconds to compute the calibrated projection matrix on a single CPU.

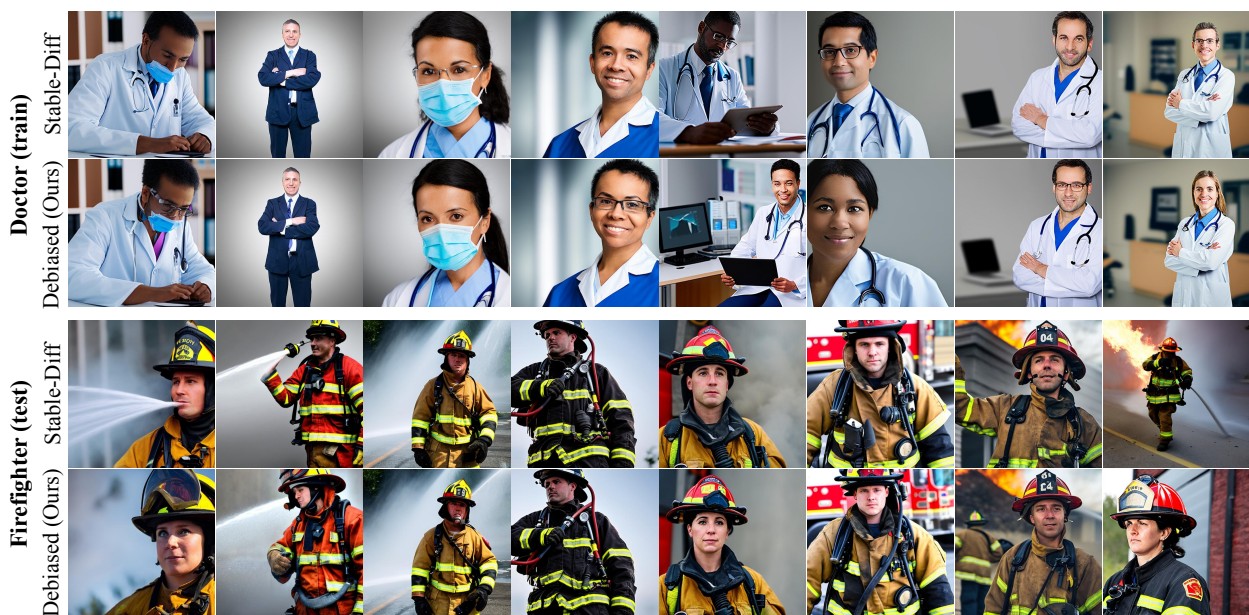

**Figure 2: Improving Gender Diversity of Stable Diffusion.** We fix the random seed of initial latent noise of Stable Diffusion (Rombach et al., 2022) and generate the images with the training / testing prompt "a photo of a doctor / firefighter". The results demonstrate that applying the calibration matrix to the prompt embedding improves the balance between male and female in the generated images.

## 7.1 Measuring the Generalization of Calibration

We first examine whether minimizing the calibration loss with training prompts can yield a calibration matrix that also works for unseen (testing) professions. In particular, we measure the average L2 difference between the projected embedding $\sum_{(i,j) \in \mathcal{S}_{\text{test}}} \|Pz_i - Pz_j\| / |\mathcal{S}_{\text{test}}|$ for testing prompts and show the results in Table 6. We can see that the calibration matrix successfully minimizes the difference after projection, even for unseen professions.

|  | Gender | | Race | |
|---|---|---|---|---|
|  | before | after | before | after |
| train | 0.56 | 0.14 | 0.70 | 0.23 |
| test | 0.54 | 0.16 | 0.69 | 0.26 |

**Table 6:** Difference between embeddings ($\lambda = 500$).

## 7.2 Quantitative and Qualitative Results

The results presented in Table 7 demonstrate a significant reduction in both gender and race discrepancy after debiasing. Importantly, the improvements are observed for both training and testing professions, implying that the obtained debiasing matrix can generalize beyond training prompts. To further illustrate the effectiveness of our approach, we present quantitative results for mitigating gender bias in Figure 2. By applying the calibration matrix to balance the male and female directions, the gender diversity of the generated images significantly improved. Additional examples can be found in appendix D.

|  |  | Gender | Race |
|---|---|---|---|
| Train | SD | 0.472±0.225 | 0.485±0.160 |
|  | Ours | **0.395±0.205** | **0.434±0.163** |
| Test | SD | 0.412±0.255 | 0.528±0.184 |
|  | Ours | **0.354±0.253** | **0.455±0.169** |

**Table 7: Discrepancy between Groups.** Calibration matrix reduces the discrepancy over gender and race. The calibration matrix derived from the training set generalizes well to testing set.

Compared to gender bias, we found that addressing racial bias is a more challenging task. One source of complexity is the ambiguity of ethnicity, as individuals may identify with multiple races. Nevertheless, as Figure 3 and Table 7 demonstrate, the diversity in the output images is improved by simply debiasing the prompt embedding with the calibration matrix.

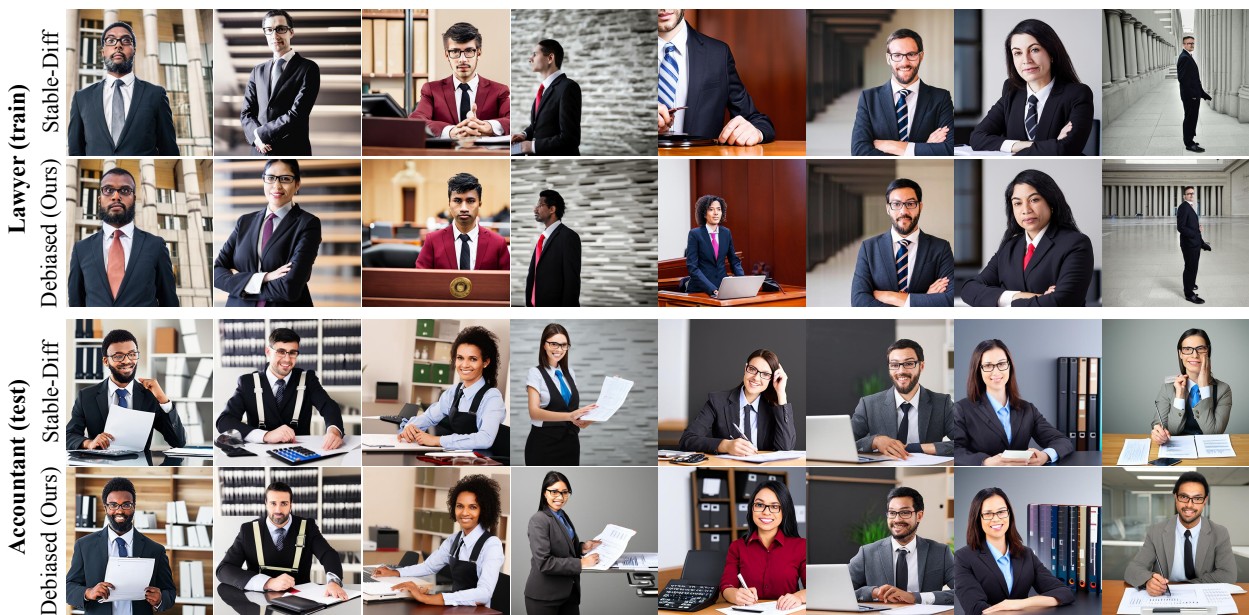

**Figure 3: Improving Racial Diversity of Stable Diffusion.** We again generate the images with Stable Diffusion. After applying the calibration matrix, the race attributes are more diverse in the generated images.

### 7.3 Human Evaluation

Despite the scalability, the prediction from a trained classifier could be erroneous. Therefore, we also evaluate our approach with human evaluation, where we invite annotators of different genders, races, and nationalities to label the sensitive attributes of the generated images. Details and the interface are included in appendix C.2. For human evaluation, we generate 25 images for each test profession, resulting in 500 images for each model. As Table 8 shows, our approach greatly improves the diversity of the generated images, namely, reducing the discrepancy between the ratio of groups.

|  | **Gender** | **Race** |
|---|---|---|
| SD | 0.472±0.257 | 0.723±0.185 |
| Ours | **0.372±0.253** | **0.589±0.188** |

**Table 8: Human Evaluation on Group Discrepancy.** We calculate the discrepancy on testing professions with human annotations. Our approach improves the diversity of Stable Diffusion by a non-trivial margin.

### 7.4 Beyond Social Biases

Our approach can also be applied to address general spurious attributes beyond social biases. As an example, we draw inspiration from the WaterBird dataset (Sagawa et al., 2019) and debias the prompt "a photo of a waterbird" by using {"a photo of a [animal] with water background" and "a photo of a [animal] with land background" } as positive pairs, where we construct a list of 100 names of animals with GPT-4 (OpenAI, 2023). As Figure 4 illustrates, our approach successfully generates images of water birds in both land and water backgrounds, whereas the original models only generated images with water background.

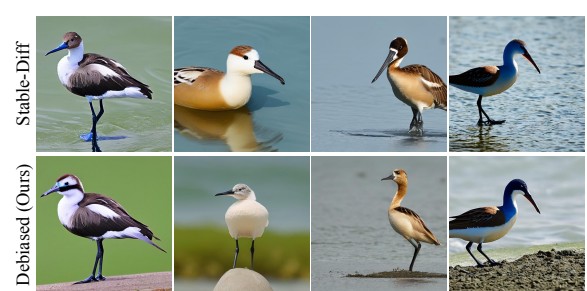

**Figure 4: Debiasing generation against Non-social Biases.** The results demonstrate the ability of the proposed method to generate images of waterbirds in both land and water backgrounds.

## 8 Conclusion

In this work, we present a new approach to debiasing vision-language foundation models by utilizing prompts to mitigate biases. The proposed calibrated projection effectively mitigates biases in both discriminative and generative vision-language models without any additional training or data. A potential area for future exploration involves the application of our technique to concept removal. This would allow for the use of our projection methods in eradicating copyrighted and inappropriate content in text-to-image models. Furthermore, the outlined objective can be expanded to accommodate non-linear scenarios, addressing more intricate biases in the model, by employing alternative optimization methods like stochastic gradient descent.

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
