# OpenReview forum: "Debiasing Vision-Language Models via Biased Prompts"
_TMLR — Rejected by TMLR_

### Review · Reviewer_RBSJ · 2023-11-03

**Summary Of Contributions:**

The paper proposes a general approach for debiasing vision-language foundation models by projecting out biased directions in the text embeddings. This approach is effective in reducing social bias and spurious correlation in both discriminative and generative vision-language models without the need for additional data or training.

In this approach, they introduce a calibration loss that minimizes the discrepancy of a pair of prompt embeddings (e.g., "photo of a male doctor" and "photo of a female doctor") after removing the bias using the projection matrix. This calibration loss has an easily solvable closed-form solution, making it suitable for large-scale models.

The proposed approach improves the group robustness of zero-shot models on well-established benchmarks and enhances the diversity of generated images from text-to-image models without altering the model parameters.

**Audience:**

Yes

**Broader Impact Concerns:**

I think it would be better if the authors added a clear ethical statement in their paper, stating the limitation of their work in debiasing the vision-language models.

**Claims And Evidence:**

Yes

**Requested Changes:**

I think it would be better if the authors added a clear ethical statement in their paper, stating the limitation of their work in debiasing the vision-language models.

**Strengths And Weaknesses:**

Strengths:

- The calibrated projection method can be applied to mitigate biases in both discriminative and generative vision-language models, making it a versatile technique for addressing biases in various applications.

- The approach is efficient and can be easily integrated into large-scale pipelines, enabling its practical implementation in real-world scenarios

- The paper is written well, and the main claims are supported by quantitative and qualitative experiments.

Weaknesses:
- This approach works well if the biases are simple and can be mitigated by linear projections.
- The choice of using discrete bias classes in this work may oversimplify the complex social constructs of biases like gender and race, potentially limiting the generalizability of the findings.
- The paper focuses on debiasing vision-language foundation models by projecting out biased directions in the text embedding, but it does not address the potential biases present in the visual embeddings or the interaction between the visual and textual modalities.
- The paper does not provide an extensive analysis of the limitations and potential ethical considerations associated with the proposed debiasing approach.

---

### Review · Reviewer_ngtZ · 2023-11-23

**Summary Of Contributions:**

This paper proposes a simple strategy to debias vision language models using biased prompts. The strategy essentially involves removing the biased components from the text embeddings using a linear projection matrix. The projection matrix essentially removes the components of the text embedding belonging to the subspace of the biased embeddings. The projection matrix is also enhanced using an auxiliary loss function on a dataset consisting of pairs of spurious prompts. This approach does not require any fine tuning of the main model and hence computationally cheap. The final projection matrix can be computed as a simple closed form, that is cheap to compute. The closed form is also shown to be related to an equalization loss in Lemma 4.2.

Experiments are performed for both the discriminative model setting and the generative model setting. The discriminative model setting involves training classifiers which are invariant to spurious correlations. Experimental results on several datasets show lower gap between the worst group loss and the average loss compared the baselines.

The generative model setting involves generating a diverse set of images from prompts. The approach is similar to the discriminative setting and it debiases the text embeddings using a learned projection matrix from pairs of data. Experimental results indicate better diversity in generated images after debiasing.

Overall the paper proposes a simple method to debias vision language. The methods seems to be promising at least for the experiments presented in the paper.

**Audience:**

Yes

**Broader Impact Concerns:**

None.

**Claims And Evidence:**

Yes

**Requested Changes:**

It seems that the results in table 3 improve with higher values of $\lambda$ with $\lambda = 1000$ being the best according to the results. However, it is unclear if increasing it further would improve the results even more.  Does it also imply that the second term in equation 3 is the most important part of the equation, and simply optimizing that part would suffice? Is the utility of the first term to ensure identifiability of the matrix P in low rank scenarios?

**Strengths And Weaknesses:**

Strengths:
- The paper proposes a simple way of debiasing vision language models using a dataset of biased prompts. The approach is simple to understand and also easy to compute.
- The approach is tested on a variety of tasks and the methods seems to be effective on the tasks presented in the paper.
- While linear debiasing approaches are not new, debiasing using biased prompts is an interesting idea showing promising results.

Weaknesses:
- The experiments presented in the paper are limited to addressing a few biases only. It would be interesting to understand the limits of the approach. How would the approach perform when a lot of biases need to be addressed? Debiasing for many objectives would lead to the removal of too many components from the embedding which may diminish the quality of the embeddings.
- There is no evidence that the biases appear as linear components in the embeddings. Further analysis could be done to show the evidence of the reduction in bias in the embeddings. For instance the embeddings could be projected to a relevant 2d plane for visualization before and after debiasing.
- The experimental results in tables 7 and 8 are not statistically significant. Confidence intervals are not provided for the other tables.

---

### Review · Reviewer_azvU · 2023-12-11

**Summary Of Contributions:**

The paper presents a novel, computationally- and data-efficient method for debiasing vision language models (VLM). It asserts that the bias towards a particular identity group can be projected away from the text representations of a VLM using a simple linear transformation. It further employs a calibration method based simply on a set of pairs of prompts where the relationship between the representations of both prompts in a pair must remain invariant to the projection. The authors derive a closed-form projection matrix for this operation thereby alleviating the need for any training data. They present debiasing evaluations for different Open-AI CLIP model variants for classification (including, non-societal group biases), and retrieval, and also demonstrate the effectiveness of the method for image generation by training Stable Diffusion with the debiased CLIP model.

**Audience:**

Yes

**Broader Impact Concerns:**

There is just one primary concern - Whether or not the ability of the debiased VLM to perform general image-language tasks like zero-shot classification is diminished.

**Claims And Evidence:**

No

**Requested Changes:**

1. The most important inclusion that I would request of the authors would be a table indicating the zero-shot scores on general image-text tasks such as classification and captioning after the debiasing. If the difference in performance on these downstream tasks is negligible, then the method proposed in the paper is of great value. Otherwise, it limits the method considerably since it destroys the core abilities of a VLM.

2. Data and arguments against the other weaknesses listed in the previous section would improve the strength of the work.

**Strengths And Weaknesses:**

Strengths:
1. The paper is well written. The authors organize their arguments in congruous theorems and employ diagrams when necessary.
2. It presents a closed-form, simple solution to an important problem - that of debiasing Vision-Language models. This makes it computationally efficient to apply, and data-efficient to train.
3. The method described in the paper appears to apply to multiple areas where  CLIP-like models are used making it quite impactful.

Weaknesses:
1. The paper does not indicate if the method is composable. For instance, Table 5 (the scores for retrieval) does not indicate if the model was simultaneously debiased for all three attributes or just one at a time. This must be clarified at the outset.
2. The paper asserts without sufficient evidence the following - (Quote: As such, embeddings of prompts such as “a photo of a [irrelevant attribute]" can capture these spurious features in the visual embedding :End-Quote). This assertion demands a citation of some prior work or at least, empirical evidence. It is unclear if the alignment between the text and visual embeddings sufficiently captures spurious attributes. A case in point is the fact that race-attributes are harder to debias from CLIP-like models indicating otherwise.
3. The method described only projects the text embeddings into a different space. It is not clear whether the alignment between the image embeddings and the text embeddings is maintained after the projection. The numbers for zero-shot classification (esp. Table 2) show that it does. But this observation is neither clarified nor stated explicitly. The paper lacks experiments indicating that this alignment between the image and text features is preserved.
4. The paper presents results for (group-representation-based) discrepancy measurement in image generation. It is not clear, however, if the discrepancy is measured using a debiased CLIP model, or with the original model. The numbers in Table 7 may be inflated (over the actual ones) if the original model was used which would fail to recognize certain identity groups owing to its bias. On the other hand, if the debiased classifier was used to measure the discrepancy, then an evaluation of the debiased model on general classification (e.g. Imagenet) must be included to indicate its capabilities.
5. Most diffusion-based models only employ text embedding as a means of conditioning (and not as an alignment loss). It is not clear if only a Vision-Language model can be employed for it. For instance, the base diffusion model of eDiff-I [Balaji, et al.] uses a T5 model which is just a language model. Therefore, the applicability of a debiased model from the diffusion standpoint may be construed as the application of a debiased Language model and not necessarily a VLM. This weakens the argument for the proposed method in this paper. If the authors wish to contradict this, a reference to prior work is required.

---

### Decision · Action_Editor_No6R · 2024-01-28

**Recommendation:** Reject

**Comment:**

Although reviewers noticed that the method is simple, novel and effective, reviewers expressed that their concerns remain. One major concern expressed by a reviewer is that it is unclear whether, after the debiasing projection, the activation vectors for image and text pair continue to align. It is unclear whether the effectiveness of the model on general image-text tasks is hurt by the debiasing approach. Another reviewer raised the point that it is unclear whether the approach can handle more complex biases. Furthermore, the paper does not provide an analysis of the limitations and potential ethical considerations of the approach.

Reviewers asked for clarification on these points but that authors didn't engage in the discussion. As such, I cannot recommend this paper for acceptance.

**Audience:**

The paper can surely be of interest of TMLR audience.

**Claims And Evidence:**

Although the paper proposes a simple and seemingly effective approach to debias vision-language models (VLM), reviewers kept some concerns about the paper, especially around the effectiveness of the resulting debiased VLM for general downstream tasks (captioning, image retrieval, etc). Authors didn't engage in the discussion with reviewers to clarify the concerns.

---

> ### Author Response · Authors · 2024-01-29
> **Request for Extension of Rebuttal Deadline**
>
> We are sincerely sorry that we missed the rebuttal deadline due to a recent transition in our professional roles, which involved changing our institutional affiliations and consequently, our primary email addresses for several authors. This change led to a delay in receiving the notification regarding the rebuttal period, and we were unable to submit our response in time.
>
> We understand the importance of the peer-review process and the role of the rebuttal in addressing reviewers' concerns. Therefore, we kindly request an extension to submit our rebuttal. We are prepared to provide a comprehensive and thoughtful response within this week, should an extension be granted. Alternatively, if an extension is not feasible, we would appreciate guidance on the possibility of resubmitting our manuscript with the necessary revisions based on the initial reviews.
>
> We sincerely apologize for any inconvenience caused by this oversight and assure you of our commitment to adhere to the revised timeline, should an extension be granted.

---

> > ### Comment · Action_Editors · 2024-01-31
> > **Unable to grant extension at this point**
> >
> > Dear Authors,
> >
> > I am sad to hear this but I cannot grant an extension at this point, given that the reviewer already made their final recommendations and the Editors approved the decision. You can still resubmit a revised version of the paper to TMLR, with the previous rounds of reviews, and what changes you made to address them: you can read the submission policy here
> >
> > https://jmlr.org/tmlr/editorial-policies.html
> >
> > -- AE